# Exploring Context Window of Large Language Models via Decomposed Positional Vectors

**Zican Dong**[1]*, **Junyi Li**[3]*, **Xin Men**[4], **Wayne Xin Zhao**[1]†, **Bingning Wang**[4]
**Zhen Tian**[1], **Weipeng Chen**[4], **Ji-Rong Wen**[1,2]
[1] Gaoling School of Artificial Intelligence, Renmin University of China
[2] School of Information, Renmin University of China
[3] Department of Computer Science, National University of Singapore
[4] Baichuan Inc.
dongzican@ruc.edu.cn, junyi_cs@nus.edu.sg
batmanfly@gmail.com, daniel@baichuan-inc.com

## Abstract

Transformer-based large language models (LLMs) typically have a limited context window, resulting in significant performance degradation when processing text beyond the length of the context window. Extensive studies have been proposed to extend the context window and achieve length extrapolation of LLMs, but there is still a lack of in-depth interpretation of these approaches. In this study, we explore the positional information within and beyond the context window for deciphering the underlying mechanism of LLMs. By using a mean-based decomposition method, we disentangle positional vectors from hidden states of LLMs and analyze their formation and effect on attention. Furthermore, when texts exceed the context window, we analyze the change of positional vectors in two settings, *i.e.,* direct extrapolation and context window extension. Based on our findings, we design two training-free context window extension methods, **positional vector replacement** and **attention window extension**. Experimental results show that our methods can effectively extend the context window length.

## 1 Introduction

Recently, Transformer-based large language models (LLMs) have demonstrated excellent capabilities on downstream tasks [1–3], in which positional encodings (*e.g.,* absolute or relative) are widely used in Transformers to better capture positional information within input sequences [4, 5]. However, LLMs typically suffer from a limited input length (called *context window*), which is constrained by the maximum length of training data. Beyond the context window, the positional encodings at larger position indices are out-of-distribution (OOD), not encountered during the training phase. Therefore, when the input sequence exceeds the context window length, there would often be a significant degradation in model performances, as evidenced by a surge in perplexity (PPL) score [6].

Prior work has primarily focused on extending the context window of existing LLMs by manipulating positional encodings. Owing to its excellent performance and long-term decay nature, RoPE [7] has been widely used to learn positional encodings for existing LLMs [8, 9]. To circumvent the OOD positional encodings in RoPE, various methods have been proposed to modify the base [10–12] or positional indices [13–16]. In addition, special relative positional encodings that apply larger negative biases to attention based on the relative distance have achieved promising length extrapolation, which

---

*Equal Contribution.
†Corresponding author.

38th Conference on Neural Information Processing Systems (NeurIPS 2024).

can effectively stabilize the model performance beyond the context window [6, 17, 18]. Furthermore, decoder-only Transformers without positional encodings (NoPE) have been found to be capable of learning implicit positional information [19], and their context window size can be extended via the adjustment of temperature hyper-parameters [20]. However, the above extension methods solely focus on adapting positional encodings or attention scores, lacking a detailed analysis of the underlying mechanisms of hidden states in LLMs.

In this work, we aim to investigate the inner working mechanism of LLMs within and beyond the context window to interpret these context window extension approaches. As the basis, our work is developed by analyzing the positional information implicitly encoded in the hidden states of LLMs across various layers and positions, both within and outside the context window. Inspired by previous work [21], we use a mean-based decomposition approach to disentangle **positional vectors** from the hidden states, which captures the information independent of semantics but related to positions.

Specifically, we first investigate how positional information is formed and examine its impact on the attention mechanism within the context window. Second, for inputs beyond the context window, we analyze the change of positional vectors in two settings, *i.e.,* direct extrapolation and context window extension. Our key findings include: (1) After the first layer, initial tokens can form distinct positional vectors, serving as anchors for shaping positional vectors in subsequent tokens; (2) Positional vectors play a critical role in modulating the long-term decay and establishing attention sinks; (3) When exceeding the context window, OOD positional vector is the major factor contributing to performance degradation, while length extrapolation can effectively keep the consistency of positional vectors both within and beyond the context window; (4) Context window extension methods enable interpolation of positional vectors by adjusting the information flow from initial tokens to subsequent tokens.

Based on the empirical findings, we further propose two training-free context window extension methods from the perspective of interpolating positional vectors: **positional vector replacement** and **attention window extension**. For LLMs with NoPE, the former method replaces the positional vectors in critical layers with interpolated ones; while for LLMs with window attention and NoPE, the latter method directly scales the window size and adjusts the temperature hyper-parameter. We evaluate the length generalization capacities of the proposed methods on PG-19 [22]. Experimental results demonstrate that our methods can effectively generalize to longer texts without fine-tuning, achieving comparable performance to previous methods.

Our main contributions are summarized as follows:

- We explicitly delineate the formation process and the effect of positional vectors, highlighting the anchoring role of initial tokens in shaping different positional vectors across tokens and their importance in achieving long-term decay and attention sinks.

- We are the first to unify length extrapolation and context window extension from the perspective of positional vectors, identifying that preventing OOD positional vectors is crucial for avoiding performance degradation.

- We propose two training-free context window extension methods via the lens of adjusting positional vectors, *i.e.,* positional vector replacement and attention window extension. Experimental results show that our methods can effectively generalize to longer texts without fine-tuning.

## 2 Background

**Transformer**  Decoder-only Transformer [4] has become the foundational architecture for LLMs [4, 8, 1]. For a Transformer with $L$ layers and a context window size $C$, given an input sequence $\mathbf{s}$ of $T$ tokens, *i.e.,* $\{x_1, \ldots, x_T\}$, we denote the output of the $l$-th layer $l$ as $\mathbf{H}_l^s = \{\mathbf{h}_{l,1}^s, \ldots, \mathbf{h}_{l,T}^s\}$. At each layer, the output $\mathbf{H}_l^s$ is obtained through multi-head attention (MHA) and feed-forward network (FFN) with residual connections applied to both components as follows:

$$\widetilde{\mathbf{H}}_l^s = \mathrm{MHA}(\mathbf{H}_{l-1}^s) + \mathbf{H}_{l-1}^s, \quad \mathbf{H}_l^s = \mathrm{FFN}(\widetilde{\mathbf{H}}_l^s) + \widetilde{\mathbf{H}}_l^s. \tag{1}$$

Finally, the output of the last layer $\mathbf{H}_L^s$ is then projected into the logits, which will be used to generate the prediction probability for each token in the vocabulary.

**Positional Vector**  Previous work has found that positional information can be learned and encoded in the hidden states of Transformers [19]. Drawing inspiration from prior work [21], we hypothesize

that each hidden state (*e.g.,* query, key, value, output of each layer) within Transformer can be decomposed into two parts, *i.e.,* a *positional vector* that captures positional information and a *semantic vector* that captures the contextual information. Taking the output $\mathbf{h}_{l,t}^s$ of the $l$-th layer at $t$-th position as an example, it can be decomposed into a positional vector $\mathbf{p}_{l,t}$ and a semantic vector $\mathbf{c}_{l,t}^s$:

$$\mathbf{h}_{l,t}^s = \mathbf{p}_{l,t} + \mathbf{c}_{l,t}^s. \tag{2}$$

Such a decomposition can disentangle two primary factors, namely positional and semantic vectors, for interpreting the internal mechanism of LLMs. Notably, since positional vectors are globally shared across different inputs, there is no superscript $s$ for $\mathbf{p}_{l,t}$. Further, the positional vector $\mathbf{p}_{l,t}$ can be decomposed into a *mean vector* $\mathbf{u}_l$ and a *positional basis* $\mathbf{m}_{l,t}$:

$$\mathbf{p}_{l,t} = \mathbf{u}_l + \mathbf{m}_{l,t}, \tag{3}$$

where the mean vector $\mathbf{u}_l$ denotes the mean of the distribution of positional vectors and the positional basis $\mathbf{m}_{l,t}$ denotes the offset of $t$-th position from the mean vector within the context window size $C$. Following previous work [21], we adopt a mean-based decomposition method to obtain the above three vectors based on $N$ samples from the training corpus as follows:

$$\mathbf{p}_{l,t} = \frac{1}{N} \sum_{s=1}^{N} \mathbf{h}_{l,t}^s, \quad \mathbf{m}_{l,t} = \mathbf{p}_{l,t} - \frac{1}{C} \sum_{t'=1}^{C} \mathbf{p}_{l,t'}, \quad \mathbf{c}_{l,t}^s = \mathbf{h}_{l,t}^s - \mathbf{p}_{l,t}. \tag{4}$$

With this decomposition, it offers an explicit way to analyze and explore the positional information encoded in the hidden states of Transformer models. For example, we can use similarity measurements to compare the positional vectors of different positions and also can visualize them in low-dimensional embedding space. In the following sections, we will mainly focus on studying the formation and impact of the positional vector $\mathbf{p}_{l,t}$, and conduct the analysis experiments.

## 3 Empirical Analysis

### 3.1 Experimental Settings

To better analyze positional information, we consider model variants with different positional encodings (PE) and attention mechanisms: variants without positional encodings (NoPE) [19] as well as variants with two different positional encodings: RoPE [7] and ALiBi [6]. We continually pre-train the TinyLlama-1.1B checkpoint [23] on 50B tokens from RedPajama [24] with a context window $C = 2048$, resulting in a set of comparison models with different positional encodings and attention mechanisms, as shown in the Table 1. *Full attention* means that each token can attend to all previous tokens, while *window attention* restricts each token to attend only to previous tokens within a window size $W$. The training details are described in Appendix A. We also evaluate common LLMs (*e.g.,* Llama-3-8B) and LLMs without positional encodings trained from scratch, and the evaluated results are listed in Appendix F.

Specifically, we subsample 32K samples with the same number of tokens from RedPajama. We perform the inference on these data to obtain hidden states of LLMs. By using the mean-based decomposition method (Eq. 4), we can obtain the positional vectors $\mathbf{p}_{l,t}$ of tokens in these sample texts.

Table 1: The compared model variants. Full attention is denoted as *Full* and window attention with a window size of $W$ tokens is denoted as *Window (W)*. We abbreviate TinyLLaMA as *TL*.

| Model | TL-NoPE | TL-RoPE | TL-ALiBi | TL-Window | TL-Window-80 | TL-Window-RoPE |
|---|---|---|---|---|---|---|
| PE | NoPE | RoPE | ALiBi | NoPE | NoPE | RoPE |
| Attention | Full | Full | Full | Window (512) | Window (80) | Window (512) |

### 3.2 Formation and Effect of Positional Vectors within Context Window

In existing LLMs, the bottom (first) layer typically takes as input token embeddings that lack inherent positional information; while interestingly, the hidden states from top layers can implicitly capture positional information, even without explicit positional encodings [19, 21, 14]. In order to have a deep understanding of implicit positional information, we next investigate the formation and effect of positional vectors in Transformers, with both full attention and window attention.

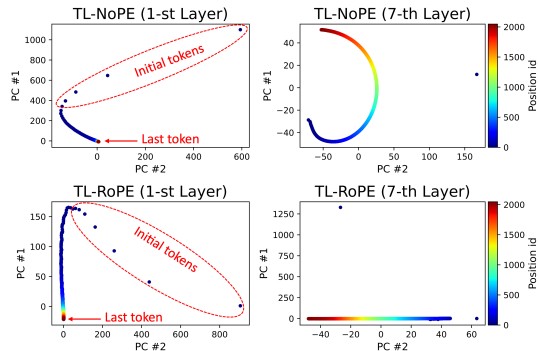
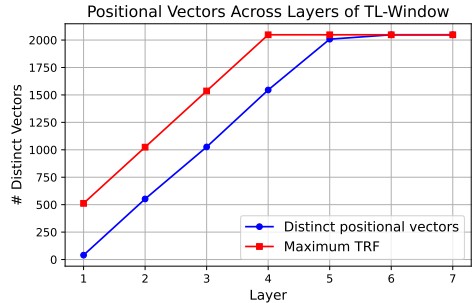

Figure 1: PCA visualization of positional vectors from the 1-st and 7-th layers.

Figure 2: Comparison of distinct positional vectors and theoretical receptive field.

### 3.2.1 Formation of Positional Vectors with Full Attention

**Positional Vector After the First Layer**    To study how positional information is distributed over different positions, we first visualize the positional vectors $\mathbf{p}_{1,t}$ (Eq. 4) decomposed from the outputs of the first layer using principal component analysis (PCA). As shown in Figure 1 (left column), initial tokens (*e.g.,* $\leq 4$ tokens) exhibit significantly distinct positional vectors, while the positional vectors of subsequent tokens are similar to each other. As a comparison, we also present the PCA results of all positional vectors at the 7-th layer. Interestingly, position vectors are evenly distributed across all the positions in Figure 1 (right column). Such a finding indicates that position vectors have captured the corresponding positional information since these vectors are distinct from each other across positions. In other words, *being distinct* can be considered as a kind of positional evidence. By comparing the left and right columns of Figure 1, it seems that only initial tokens are different from the rest tokens after the first layer, which might suggest that **after the first layer, initial tokens have already formed distinct positional information but subsequent tokens have not yet established such information.**    To investigate the reasons behind this phenomenon, we select the first attention head in the first layer (similar to other heads) to analyze attention scores, as detailed in Appendix B. We can prove that the positional vector $\mathbf{p}_{1,1}$ for the first token is different from the following tokens and the attention scores affect the formation of positional information. Thus, through several layers, the tokens after the first token will gradually form distinct positional vectors (Figure 1 right column).

**Positional Information Flow From Initial Tokens**    By applying positional vectors at top layers (PCA visualized in Appendix G), we find that after forwarding several layers, tokens at all positions can also exhibit distinct positional vectors, and similar findings are also found in previous work [19]. To trace back to the source of positional information, a reasonable speculation is that initial tokens play a key role in the formation of positional information for the rest tokens since only initial tokens capture positional information after the first layer. To validate this, we select two groups of reference tokens: initial tokens (1∼4) and secondary tokens (4∼256), and further analyze what information of these tokens is critical for the formation of positional information in subsequent tokens (>256). Thus, based on a top-down strategy, we conduct an ablation study for each group by respectively deleting the value $\mathbf{v}_{l,t}^{s}$ of attention module (*w/o value*), the semantic vector $\mathbf{c}_{l,t}^{s}$ (*w/o semantic vector*), positional vector $\mathbf{p}_{l,t}$ (*w/o positional vector*), and positional basis $\mathbf{m}_{l,t}$ (*w/o positional basis*). Then, for each variant, we average the outputs of all layers and decompose new positional vectors based on Eq. 4. Finally, we compute the average cosine similarity between the original and new positional vectors for those subsequent tokens (>256) and also report PPL on samples in RedPajama. From Table 2, we can see that removing the positional vector and basis of 1∼4 tokens largely affect the positional vectors at later positions (low similarity). Conversely, removing the semantic vector or altering secondary tokens has slight effects on both similarity and PPL. From these findings, we conclude that **the positional vectors of initial tokens seem to serve as the role of anchors, largely contributing to the formation of positional information in subsequent tokens.**

Table 2: Results of removing different components in attention. *Sim* denotes the cosine similarity between original and new positional vectors, and *PPL* denotes perplexity on RedPajama.

| | | original | w/o value | | w/o positional vector | | w/o positional basis | | w/o semantic vector | |
|---|---|---|---|---|---|---|---|---|---|---|
| | | - | 1~4 | 4~256 | 1~4 | 4~256 | 1~4 | 4~256 | 1~4 | 4~256 |
| TL-NoPE | Sim | 1 | -0.1558 | 0.9797 | -0.1810 | 0.9086 | -0.1817 | 0.9046 | 0.9985 | 0.9514 |
| | PPL | 7.56 | >1000 | 8.97 | >1000 | 13.36 | >1000 | 10.23 | 8.20 | 10.55 |
| TL-RoPE | Sim | 1 | 0.8394 | 0.9902 | 0.8505 | 0.9874 | 0.1711 | 0.9944 | 0.9970 | 0.9596 |
| | PPL | 6.06 | 11.98 | 6.44 | 12.28 | 6.24 | >1000 | 6.11 | 6.63 | 6.85 |

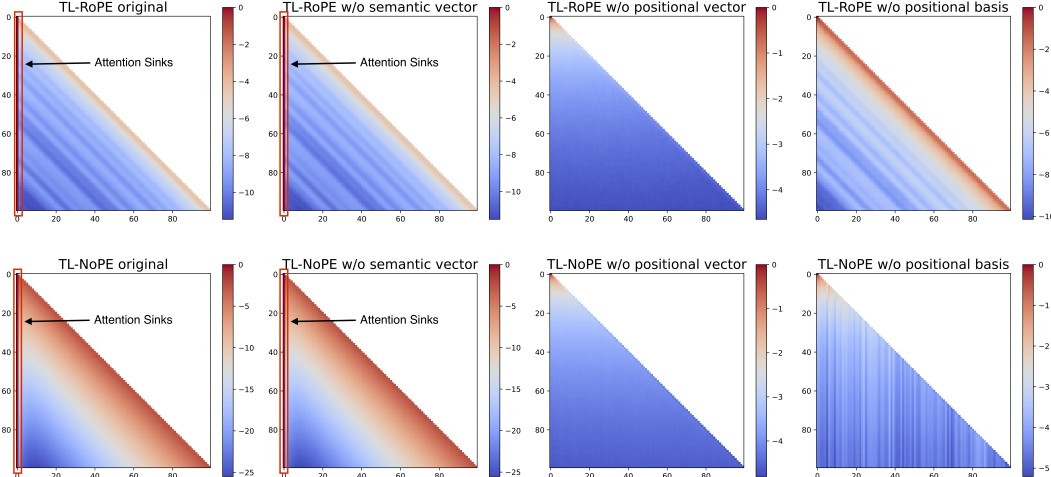

Figure 3: Logarithmic attention maps of TL-RoPE, and TL-NoPE.

### 3.2.2 Formation of Positional Vectors with Window Attention

Unlike full attention, LLMs employing window attention restrict each token to attend only to tokens within a window size. Previous work has shown that the maximum theoretical receptive field (TRF) in window attention is equal to the product of the window size $W$ and the layer index $l$ [18].

To analyze how positional vectors change across layers, we compute *the number of distinct positional vectors* within the maximum TRF. Notably, those tokens beyond the maximum TRF share the same positional vectors due to translation invariance [18]. Specifically, we first randomly select a positional vector outside the maximum TRF and then compute the cosine similarity between positional vectors within the maximum TRF and the selected vector. We consider the positional vector with a similarity score lower than a threshold (*i.e.,* 0.99) as *distinct*. Figure 2 presents the number of distinct positional vectors and TRF at each layer. We can see that after the first layer, only initial tokens show distinct positional vectors, further verifying the findings in Section 3.2.1. As the layer increases, more tokens display different positional information and the number of distinct positional vectors increases by 512 (a window size $W$) with each additional layer. The reason is that due to the constraint of window attention, each token at the preceding layer can only influence tokens within the window size at the next layer. As *being distinct* indicates the formation of position information, **similar positional information flow from initial tokens to subsequent tokens also occurs for window attention, but gradually propagating across both windows and layers**.

### 3.2.3 Effect of Positional Vectors on Attention

After discussing the formation of positional vectors, we explore their impact on the attention module, mainly focusing on the attention scores. We first extract queries and keys from each head in all layers, and then compute the average attention scores in the following four settings, including (1) *original:* the original model, (2) *w/o semantic vector:* removing the semantic vectors of keys and queries, (3) *w/o positional vector:* removing the positional vectors of keys and queries, (4) *w/o positional basis:* removing the positional basis of keys and queries. Figure 3 presents the logarithmic attention scores for the first 100 tokens in the first head and fifth layer (similar results in many other heads and layers).

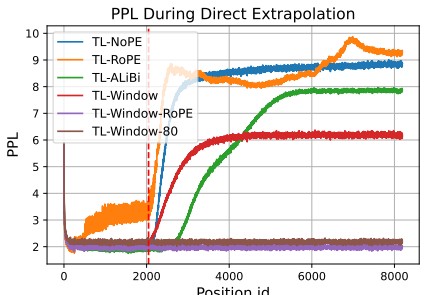
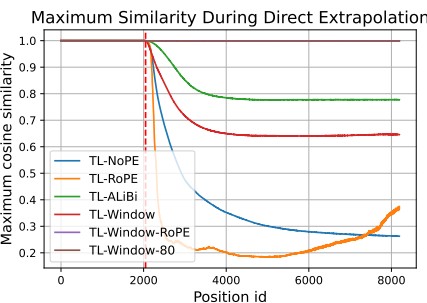

Figure 4: **Left**: The average PPL across positions during direct extrapolation. **Right**: The maximum cosine similarity between positional vectors within and beyond context window during extrapolation.

**Effect of Positional Vectors on Attention Sinks**    Previous work has found that the initial tokens will be assigned high attention scores, called "*attention sinks*" [15], which can be clearly observed in Figure 3. However, once the positional vector or positional basis is removed from the keys and queries, the attention scores between initial tokens and other tokens drop significantly for TL-NoPE and TL-RoPE. This finding suggests that **the presence of attention sinks is likely attributed to the inherent positional information in the positional vectors of initial tokens.**

**Effect of Positional Vectors on Long-term Decay**    For long texts, the attention scores of LLMs often exhibit a long-term decay pattern, which means that the score decreases as the relative distance between tokens increases [7, 6]. However, as shown in Figure 3, when removing the positional vector or positional basis, TL-NoPE fails to exhibit long-term decay. Even with explicit relative positional encoding, the distribution of attention scores in TL-RoPE tends to be smooth after removing decomposed positional vectors. Therefore, **positional vectors also play a crucial role in the long-term decay property of attention scores.**

### 3.3    Effect of Positional Vectors beyond Context Window

Typically, when dealing with texts that exceed the context window, there are two lines of research, *i.e.,* direct extrapolation and context window extension. In this section, we aim to investigate the change of positional vectors in these two methods for revealing their effectiveness.

#### 3.3.1    Direct Extrapolation

**Relationship Between Positional Vectors and Length Extrapolation Ability**    To examine the impact of positional vectors in direct extrapolation, we reuse the trained model variants in Table 1 to perform inference on samples consisting of 8192 tokens. Further, we analyze the change in PPL score and the maximum cosine similarity between positional vectors within and beyond the context window. As shown in Figure 4 Left, only TL-Window-RoPE and TL-Window-80 demonstrate the length extrapolation ability, maintaining stable PPL across longer texts. These models can preserve the consistency of positional vectors both within and beyond the context window (high similarity in Figure 4 Right). Conversely, the rest models, including those with extrapolated positional encodings or window attention (*e.g.,* TL-ALiBi), struggle to generalize to longer contexts. Notably, these models exhibit rapid changes in positional vectors (beyond 2048), diverging from the distributions observed within the context window. Thus, our findings underscore **the critical role of the stability of positional vectors in enhancing the capability for length extrapolation.**

**Effect of OOD Positional Vectors**    Beyond the context window, position vectors are not encountered during training and are out-of-distribution from those vectors within the context window. To explore whether OOD positional vector is a key factor in performance degradation, we select TL-NoPE for evaluation, which does not use explicit positional encodings. First, we compare the attention distribution within and beyond the context window. Figure 5 shows the attention map and scores between initial and rest tokens by averaging all heads of the 5-th layer (similar results in other layers). Once exceeding the context window ($T = 2048$), the attention distribution in these positions changes sharply, losing the characteristics of attention sinks and long-term decay. Since these properties

highly depend on the positional vectors within the context window, we speculate that **OOD positional vectors disrupt the original attention distribution**. Besides, we feed the positional vectors of the last layer into the linear projection of the softmax layer to get the logits at different positions. Figure 5 (Right) presents that the logits within the context window are tightly similar while others show different distributions. Thus, **the OOD positional vectors can damage the token prediction probability distribution**, thereby leading to performance degradation.

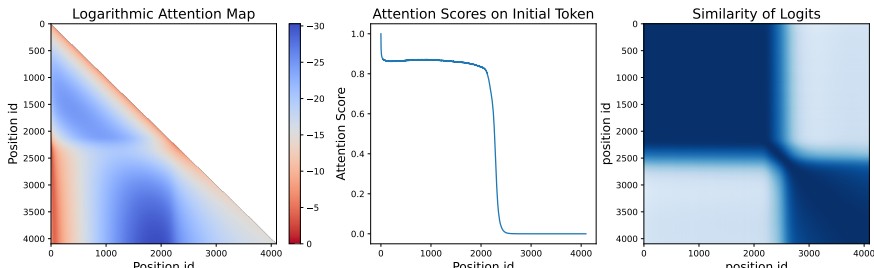

Figure 5: **Left**: Attention map of TL-NoPE. **Middle**: Attention Scores between initial token and others in TL-NoPE. **Right**: Similarity of logits of positional vectors across positions in TL-NoPE.

Table 3: The interpolation results of positional vectors, where *Factor* ($=$ Target Length$/C$) is the expansion factor of the context window, *Ratio* is the effective interpolation ratio of positional vectors (detailed in Appendix C), and *Similarity* is the average cosine similarity between the scaled positional vector and the original most similar positional vector by averaging all layers.

| Model | Method | Target Length | Factor | Ratio | Similarity | PPL/$\Delta$PPL |
|---|---|---|---|---|---|---|
| TL-NoPE | Attention Scaling ($\lambda = 1.2$) | 4096 | 2 | 2.56 | 0.98 | 8.95/+1.42 |
| | Attention Scaling ($\lambda = 1.3$) | 8192 | 4 | 4.30 | 0.94 | 17.87/+10.34 |
| | Initial Scaling ($\lambda = 1.2$) | 4096 | 2 | 2.38 | 0.97 | 9.82/+2.29 |
| | Initial Scaling ($\lambda = 1.3$) | 8192 | 4 | 4.10 | 0.91 | 32.78/+25.25 |
| TL-RoPE | Dynamic NTK | 4096 | 2 | 2.05 | 0.99 | 6.00/-0.02 |
| | Dynamic NTK | 8192 | 4 | 3.75 | 0.96 | 6.78/+0.76 |

### 3.3.2 Context Window Extension

**Change of Positional Vectors When Extending Context Windows**     To investigate why context window extension can prevent performance degradation, we analyze the change of positional vectors in two training-free context window extension methods, including dynamic-NTK [11] for TL-RoPE and attention scaling ($\mathbf{q}_i \mathbf{k}_j$ multiplied by a scaling factor $\lambda$) [20] for TL-NoPE. From Figure 6, we can see that after context window extension, positional vectors have undergone interpolation compared to the original ones. Comparing the *Factor* and *Ratio* metrics in Table 3, we conclude that **the effective interpolation ratio is close to the expansion factor** (*e.g.,* 2 vs 2.56). Besides, as the expansion factor increases, there is a decrease in *Similarity* and an increase in *PPL*. Therefore, we suspect that imperfect interpolation may be a major reason for the decline in model performance.

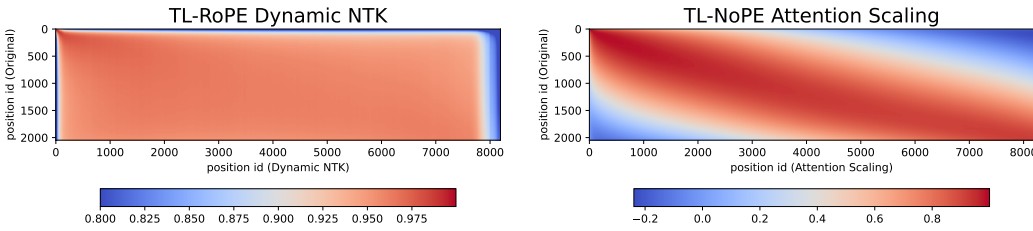

Figure 6: The average cosine similarity between the scaled and original positional vectors.

**Effect of Initial Tokens on Context Window Extension**    Since the initial tokens serve as the anchor for the formation of subsequent positional vectors, we evaluate whether changing the information flow from the initial tokens to the rest tokens can achieve the interpolation effect. To avoid the effect of OOD positional encodings, we follow the attention scaling method on TL-NoPE but only scale the attention logits between the initial tokens and others, denoted as *Initial Scaling*. As shown in Table 3, it can achieve comparable performance and interpolation ratios closer than scaling all attention logits in Attention Scaling (*e.g.,* 2.38 vs 2.56), further underscoring that **the interpolation of positional vectors is mainly achieved by adjusting the information flow of anchor tokens.**

## 4    Extending Context Window via Positional Vectors

Inspired by our analysis of the formation of positional vectors and the interpolation of positional vectors when extending the context window, we propose two training-free context window extension methods, *i.e.,* **positional vector replacement** and **attention window extension**. The pseudocode of these methods can presented in Appendix E.

### 4.1    Positional Vector Replacement

In Section 3.2.3, we show that when exceeding the context window, the OOD positional vectors tend to cause the collapse of attention distribution. Further, we observe that context window extension methods can achieve length interpolation of positional vectors and the effective interpolation ratio is close to the expansion factor of the context window. Thus, we propose to replace all the implicitly learned positional vectors with the interpolated ones, called *positional vector replacement*, to avoid the OOD issue in LLMs without positional encodings (NoPE).

Specifically, we linearly interpolate the positional vectors within the context window with an interpolation ratio $r$ and multiply the interpolated ones with a times $\alpha$ ($\geq 1$). In practice, we find that properly increasing the interpolation ratio $r$ and times $\alpha$ can achieve better effectiveness of interpolation (details are discussed in Appendix D). Owing to the critical role of initial tokes, the positional vectors of the first four tokens remain unchanged, while those of subsequent tokens are replaced with the interpolated vectors. The replaced output $\hat{\mathbf{h}}_{l,t}$ for each layer can be formulated as:

$$\hat{\mathbf{h}}_{l,t} \quad = \quad \mathbf{h}_{l,t} - \mathbf{p}_{l,t} + \alpha\hat{\mathbf{p}}_{l,t}, \tag{5}$$

$$\{\hat{\mathbf{p}}_{l,5}, \ldots, \hat{\mathbf{p}}_{l,r(C-4)+5}\} \quad = \quad \text{Interpolation}(\{\mathbf{p}_{l,5}, \ldots, \mathbf{p}_{l,C}\}), \tag{6}$$

where $C$, $l$, and $s$ represent the original context window size, replaced layer, and interpolation ratio. Since replacing positional vectors for all layers requires heavy recalculation efforts and the positional information is passed across layers, we only apply the replacement strategy to a single early layer. We find that the 4-th layer is the optimal layer for replacement in TL-NoPE, as shown in Figure 8.

### 4.2    Attention Window Extension

As discussed in Section 3.2.2, the positional vectors are shaped across layers and windows by the distinct positional information of initial tokens. Inspired by these observations, we propose *attention window extension*, the first training-free length interpolation method for window attention-based LLMs without positional encodings. The core idea is to extend the attention window size to control the formation of positional vectors. When scaling the context window by a ratio, the window size also needs to be extended by the same interpolation ratio $r$. However, for positions in the extended first window $\{W + 1, \ldots, rW\}$, their position vectors are OOD. To avoid this, we follow the attention scaling method [20] and scale the attention logits with a scaling factor $\lambda$, achieving better interpolation of positional vectors. We define the attention score $a_{ij}$ between query $\mathbf{q}_i \in \mathbb{R}^{D_H}$ and key $\mathbf{k}_j \in \mathbb{R}^{D_H}$ for any heads and layers as:

$$a_{ij} = \frac{\exp(\lambda\mathbf{q}_i\mathbf{k}_j/\sqrt{D_H})}{\sum_{z=i-rW}^{i} \exp(\lambda\mathbf{q}_i\mathbf{k}_z/\sqrt{D_H})}. \tag{7}$$

### 4.3    Results on Language Modeling

To assess the effectiveness of our proposed methods, we evaluate language modeling performance on the test set of PG-19 [22]. In line with previous work [6], we measure PPL across various input

Table 4: Results of language modeling in PG-19. The context window size $C$ is 2048.

| Model | Interpolation Method | Factor | 2K | 4K | 6K | 8K |
|---|---|---|---|---|---|---|
| TL-RoPE | - | - | 10.17 | $> 10^3$ | $> 10^3$ | $> 10^3$ |
| | Dynamic NTK | - | 10.17 | 10.45 | 11.28 | 28.58 |
| TL-NoPE | - | - | 11.92 | $> 10^3$ | $> 10^3$ | $> 10^3$ |
| | Attention Scaling | $\lambda = 1.2$ | 17.03 | 17.05 | 54.26 | $> 10^3$ |
| | | $\lambda = 1.3$ | 32.07 | 43.84 | 51.50 | 46.59 |
| | Positional Vector Replacement (ours) | $r = 2, \alpha = 1.1$ | 13.54 | 15.58 | - | - |
| | | $r = 5, \alpha = 1.3$ | 28.15 | 47.65 | 49.79 | 73.79 |
| TL-Window | - | - | 12.86 | 713.51 | 660.30 | 660.51 |
| | Attention Window Extension (ours) | $r = 2, \lambda = 1.1$ | 13.70 | 14.10 | - | - |
| | | $r = 4, \lambda = 1.2$ | 17.23 | 31.66 | 29.27 | 29.30 |

lengths (from 2K to 8K) using a sliding window approach. We apply positional vector replacement to TL-NoPE and attention window extension to TL-Window. All the hyper-parameters are selected according to the PPL and the change of positional vectors across layers. For compared baselines, we select Dynamic-NTK [11] for TL-RoPE and Attention Scaling [20] for TL-NoPE.

The results are shown in Table 4. First, without interpolation, the PPL increases extremely after beyond the context window (*e.g.,* $> 10^3$). When using the positional vector replacement or attention window extension methods, we observe that PPL decreases substantially, showing the effectiveness of our proposed methods. Compared to attention scaling, our attention window extension method successfully extends the context window to 8K tokens with lower PPL. Moreover, our positional vector replacement method achieves similar performance to attention scaling within 6K tokens but shows increased PPL at 8K. We attribute this phenomenon to the decreasing effective interpolation ratio across layers, as shown in Figure 9. Additionally, an increase in PPL with the rising interpolation ratio $r$ is also observed in both our methods, likely due to imperfect interpolation of positional vectors.

## 5 Related Work

**Position Information in Transformers**  Positional information was crucial in Transformer-based LLMs, to enhance the sequence modeling abilities. The vanilla Transformer introduced absolute positional encodings, using a unique embedding to each position and adding it to the corresponding input embedding [4]. In contrast, relative positional encodings introduced biases based on the relative distance between tokens within attention modules [25–27, 6, 7]. Besides explicit positional encodings, some work investigated the implicit positional information within hidden states of Transformers. Even without positional encodings, positional information was found in hidden states of Transformer decoders [19, 28, 29]. Besides, prior work decoupled positional basis from hidden states in Transformers and analyzed geometric properties [21]. Our work mainly explores positional information embedded in the hidden states of LLMs, examining the formation and impact of positional vectors, and using it to analyze the mechanism of context window for LLMs.

**Extending Context Window**  LLMs were often constrained by pre-defined context windows. When processing inputs that exceed these windows, models typically encountered OOD issues, leading to significant performance degradation. To meet the growing demands of long context tasks [30, 31], various methods were proposed to address this limitation and model longer texts, which can be roughly categorized into length extrapolation and context window extension [32]. Length extrapolation techniques aimed to maintain stable PPL regardless of text length by designing specialized positional encodings or window attention mechanisms [6, 17, 18, 15, 14]. Conversely, context window extension methods focused on extending the context window of existing models by adapting positional encodings or temperature hyper-parameters, thereby enlarging the context window with minimal performance loss [13, 12, 16, 20, 11, 10]. This paper bridges the concepts of length extrapolation and context window extension through the lens of positional vectors, enhancing the interpretability of context windows in LLMs.

# 6 Conclusion

In this work, we explored the inner working mechanism of LLMs within and beyond the context window via decomposed positional vectors. We found that the initial tokens initially present different positional information and serve as anchors for shaping the positional vectors of subsequent tokens. Besides, after exceeding the context window, length extrapolation methods maintain the stability of positional vectors, while context window extension methods achieve the interpolation of positional vectors. Based on our observations, we proposed two methods: positional vector replacement and attention window extension, which achieve training-free context window extension for specific LLMs. We believe that positional vectors will serve as an effective tool for analyzing the context window of LLMs and promote the design of better algorithms for extending the context windows of LLMs.

# 7 Limitation

Our work provides an extensive discussion and analysis of the context window through the lens of positional vectors. However, our study is mainly constrained by the use of small-scale LLMs that we trained ourselves, due to the unavailability of existing LLMs with the specific positional encodings and attention patterns required for our experiments. Though some mainstream LLMs are evaluated, these models are all based on RoPE. Furthermore, we have demonstrated the effectiveness of our proposed methods solely on our own models, again limited by the absence of suitable external models. In future work, we aim to seek a broader range of models to validate our findings more comprehensively.

## Acknowledgement

This work was partially supported by National Natural Science Foundation of China under Grant No. 62222215, Beijing Municipal Science and Technology Project under Grant No. Z231100010323009, and Beijing Natural Science Foundation under Grant No. L233008. Xin Zhao is the corresponding author.

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

# A  Training and Experimental Details

We continue pre-training all our models from the TinyLlama[3] [23] checkpoint on the RedPajama [24] dataset. All models undergo the same training process, with differences only in their positional encoding and attention patterns. Each model is trained on 16 A800 GPUs over two days. Detailed training parameters are provided in Table 5.

Table 5: Training Details of Models.

| Training Data | RedPajama [24] |
|---|---|
| Tokens | 50B |
| Parameters | 1.3B |
| Context Window Size | 2048 |
| Decay style | cosine |
| Learning Rate | 2e-5 |
| Min Learning Rate | 1e-6 |
| Optimizer | AdamW(0.95,0.9) |
| Warmup Steps | 3000 |
| Batch size | 48 |
| Gradient clipping | 1.0 |

In addition, all the subsequent experiments are computed in 8 A800 GPUs.

# B  The Positional Vectors After the First Layer

Though previous work [29, 28] have proven that implicit positional information can be encoded in hidden states after one attention module, they only set the attention logits are equal regardless of queries and keys, which does not hold in actual Transformers. In this section, we demonstrate the preference in attention scores promotes the formation of different positional information in the initial tokens.

## B.1  Details

For the $s$-th sample in the corpus, we denote $\mathbf{v}_{1,i}^s$, $\hat{\mathbf{h}}_{1,i}^s$ as the value and output of the first attention head and the first layer at position $i$, and $a_{i,j}^s$ as the attention score between position $i$ and position $j$. We also denote $\hat{\mathbf{p}}_{1,i}$ as the positional vector of the attention output, and $\mathbf{u}_v$ as the mean vector of values. The positional vector can be represented as the formula:

$$\hat{\mathbf{p}}_{1,1} = \frac{1}{N} \sum_{s=1}^{N} \hat{\mathbf{h}}_{1,1}^s = \frac{1}{N} \sum_{s=1}^{N} \mathbf{v}_{1,1}^s = \mathbf{u}_v \tag{8}$$

$$\hat{\mathbf{p}}_{1,i} = \frac{1}{N} \sum_{s=1}^{N} \hat{\mathbf{h}}_{1,i}^s = \frac{1}{N} \sum_{s=1}^{N} \sum_{j=1}^{i} a_{i,j}^s \mathbf{v}_{1,j}^s \tag{9}$$

For the first token, the output is equal to the value of itself, so the positional vector is equal to the mean of values. In addition, When $a_{i,j}^s = 1/i$, positional information of all positions is equal as follows:

$$\hat{\mathbf{p}}_{1,i} = \frac{1}{N} \sum_{s=1}^{N} \sum_{j=1}^{i} \frac{1}{i} \mathbf{v}_{1,j}^s = \frac{1}{i} \sum_{j=1}^{i} \frac{1}{N} \sum_{s=1}^{N} \mathbf{v}_{1,j}^s = \frac{1}{i} \sum_{j=1}^{i} \mathbf{u}_v = \mathbf{u}_v. \tag{10}$$

However, due to the preferences in attention scores, values that can be decomposed into more vector $\hat{\mathbf{p}}_{1,i} - \mathbf{u}_v$ will be assigned larger weights, making the positional information of the following tokens differ from the beginning token. In summary, the preferences of attention scores force the formation of different positional vectors of the following tokens with the initial ones.

---

[3] https://huggingface.co/TinyLlama/TinyLlama-1.1B-intermediate-step-1431k-3T

## B.2 Proof of Preference in Attention Scores

Following previous work [28], we only focus on a single attention head in the first layer with specific weights. In our parameterization, we only consider the first two dimensions. We demonstrate the capacity of causal Transformers to learn this capacity with either NoPE or RoPE.

In our settings, the Transformer has $H$ attention heads in each layer and the word embedding matrix is $\mathbf{W}_E \in \mathbb{R}^{D \times V}$, where $D$ is the dimension of the hidden states and $V$ is the number of vocabulary. We set that the first dimension in word embedding conforms to normal distribution $\mathcal{N}(0,1)$, (*i.e.,* $e_{1,t} \sim \mathcal{N}(0,1), \forall t \in \{1, \dots, V\}$), while the second dimension is 1. Other dimensions are arbitrary values. Thus, the word embedding matrix $\mathbf{W}_E$ is:

$$
\mathbf{W_E} = \begin{bmatrix}
e_{1,1} & e_{1,2} & e_{1,3} & \cdots & e_{1,V} \\
1 & 1 & 1 & \cdots & 1 \\
e_{3,1} & e_{3,2} & e_{3,3} & \cdots & e_{3,V} \\
\vdots & \vdots & \vdots & \ddots & \vdots \\
e_{D,1} & e_{D,2} & e_{D,3} & \cdots & e_{D,V}
\end{bmatrix}_{D \times V}
\tag{11}
$$

Next, we set the projection matrix of query, key, and value as $\mathbf{W}_Q, \mathbf{W}_K, \mathbf{W}_V$ of the first layer and first head of Transformers.

$$
\mathbf{W}_Q = \begin{bmatrix}
0 & 1 & \cdots & 0 \\
0 & 1 & \cdots & 0 \\
\vdots & \vdots & \ddots & \vdots \\
0 & 1 & \cdots & 0
\end{bmatrix}_{\frac{D}{H} \times D},
\mathbf{W}_K = \begin{bmatrix}
1 & 0 & \cdots & 0 \\
1 & 0 & \cdots & 0 \\
\vdots & \vdots & \ddots & \vdots \\
1 & 0 & \cdots & 0
\end{bmatrix}_{\frac{D}{H} \times D},
\mathbf{W}_V = \begin{bmatrix}
1 & 0 & \cdots & 0 \\
1 & 0 & \cdots & 0 \\
\vdots & \vdots & \ddots & \vdots \\
1 & 0 & \cdots & 0
\end{bmatrix}_{\frac{D}{H} \times D}.
\tag{12}
$$

$\mathbf{W}_K$ make sure that only the first dimension will be considered in attention scores. $\mathbf{W}_Q$ transfers the second dimension in the input to the first dimension. For Transformers without positional vectors (NoPE), the attention logits for each key at position $i$ is $a_{i,j}^s = e_{1,s_i}$. Thus, the vectors with large first dimensions will be assigned with large attention logits, which proves that the model without positional vectors can learn to preference some values regardless of the query.

For Transformer with RoPE, we assume that the first two dimensions correspond to the basis of the rope with a value of $1/10000$ and the maximum context window size is 2048. Thus, for the largest relative distance $i - j$, the rotation angle is smaller than $\frac{\pi}{2}$. Thus, the attention score can be represent as $a_{i,j}^s = e_{1,s_i} \cos((i-j)/10000)$. Thus, the attention logits will be larger than zero only if the first dimension of the key is larger than zero. In addition, for the same relative distance, keys with larger first dimensions have large attention logits. Thus, keys with positive values in the first dimension will be assigned greater attention weights.

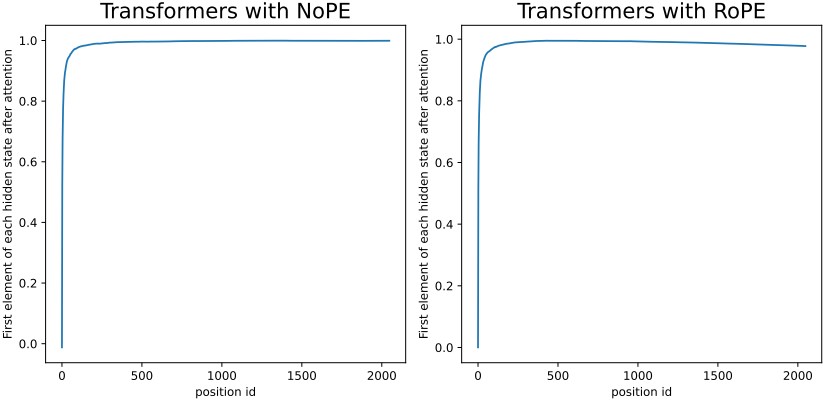

Figure 7: The values of first elements of the output of single head attention due to attention preferences in Transformers with NoPE and RoPE.

We also experimentally examine the first element of the output at different positions. With the above settings, we generate 10000 sequences with a length of 2048 from this distribution. Then,

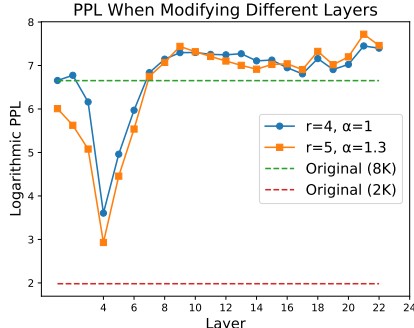
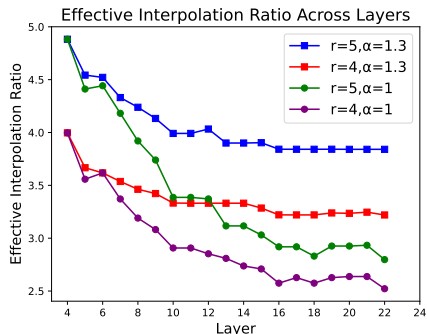

Figure 8: Changes of PPL with replacement at different layers.

Figure 9: Effective scaling factors of positional vectors at each layer.

we compute the averaged first element of each hidden state after attention in the first layer and head, which represents the positional vectors at that position for both NoPE and RoPE. As shown in Figure 7, the first element of the output increases fast at initial positions. Here, we can observe that the attention preferences make the first element of the output increase fast at the initial positions and tend to stabilize at a later position, further proving the different positional vectors of following tokens from the initial ones.

In summary, Transformers with either NoPE or RoPE can learn preference in attention score. In addition, the preference in attention scores forces the different positional vectors of the subsequent tokens with the initial tokens.

## C  Defination of Effective Interpolation Ratio

Given a Transformer with a context window size $C$. For samples with length $C'(C' > C)$, we disentangle positional vectors $\{\mathbf{p}_{l,1}, \ldots, \mathbf{p}_{l,C'}\}$. Subsequently, we employ a context window extension method and re-compute the position vectors $\{\mathbf{p}'_{l,1}, \ldots, \mathbf{p}'_{l,C'}\}$. We first define the corresponding position indices $f(\mathbf{p}'_{l,t})$ for positional vector after extension $\mathbf{p}'_{l,t}$) as the indices of the most similar positional vectors before extension:

$$f(\mathbf{p}'_{l,t}) = \arg \max_{1 \leq i \leq C'} \frac{\mathbf{p}_{l,i}^T \mathbf{p}'_{l,t}}{\|\mathbf{p}_{l,i}\| \|\mathbf{p}'_{l,t}\|}. \tag{13}$$

Then, the effective interpolation ratio $r'$ can be represented as the ratio between the maximum indices of position vectors after extension whose corresponding position indices are the context window size $C$, as shown in Equation 14.

$$r = \frac{\arg \max_{1 \leq t \leq C'} (f(\mathbf{p}'_{l,t}) = C)}{C}. \tag{14}$$

## D  Analysis of Positional Vector Replacement

### D.1  Optimal Replacement Layer

Replacing positional vectors for all layers requires heavy recalculation efforts, so we only select one critical layer to apply the replacement strategy. We evaluate the performance (logarithmic PPL score) of our replacement strategy at each layer in TL-NoPE, using different interpolation ratios and expansion factors, i.e.,$(r = 4, \alpha = 1)$ and $(r = 5, \alpha = 1.2)$, on samples with 8K tokens from RedPajama. As shown in Figure 8, the PPL is the lowest when replacing the 4-th layer of TL-NoPE. Thus, we choose the 4-th layer for replacement in TL-NoPE in other experiments.

### D.2  Effective Interpolation Ratio

We further examine the effective interpolation ratio of positional vectors with different settings. In our setting, we only replace the positional vectors in the 4-th layer of TL-NoPE with interpolated

positional vectors in four settings on samples with 8K tokens from RedPajama: (1) $r = 4, \alpha = 1$, (2) $r = 5, \alpha = 1$, (3) $r = 4, \alpha = 1.3$, (4), $r = 5, \alpha = 1.3$.

Figure 9 presents the effective interpolation ratio in each layer. As the layer increases, the effective interpolation ratio decreases as the layer increases. When the interpolation ratio is equal to the expansion factor of the context window, *i.e.,* $r = 4$, the effective interpolation ratio is unavoidably smaller than the expansion factor, leading to degraded performance. In addition, multiplying the interpolated positional vectors with a larger times *e.g.,* $\alpha = 1.3$, alleviates the decrease of effective interpolation ratio across layers. Thus, we suggest to properly increase the interpolation ratio $r$ and times $\alpha$.

# E    Pseudo Code

## E.1    Positional Vector Replacement

For Positional Vector Replacement, we give the implementation with Pytorch code in Algorithm 1, which can be inserted after the output of the selected layer.

---

**Algorithm 1** PyTorch-style Pseudocode of Positional Vector Replacement

---

```
h, p # hidden states, positional vectors
T, layer, s, alpha # context window size, interpolation ratio, selected layer, scaling factor of positional
vectors.

h[:,4:] -= p[layer, 4:h.shape[1]].unsqueeze(0)
# removing original positional vectors.

interpolated = torch.nn.functional.interpolate(p[layer, 4:T].transpose(0,1).unsqueeze(0), size
= int(T*s), mode = 'linear', align_corners=True).transpose(1,2)
# Linear interpolation of positional vectors.

h[:,4:] += alpha*interpolated[:,:h.shape[1]-4]
# Replacing with new positional vectors.
```

---

## E.2    Attention Window Extension

For attention window extension, we give the implementation with Flash-Attention-2 [33] in Algorithm 2.

---

**Algorithm 2** PyTorch-style Pseudocode of Attention Window Extension

---

```
query, key, value, attn_output # queries, keys, values, and output in the attention model
W, lambda, s # original window size, scaling factor of attention logits, window extension factor
flash_attn_varlen_func # attention function in flash attention 2.

new_window = W*s # extended window size

attn_output = flash_attn_varlen_func( query, key*lambda, value, ..., window_size = (new_window,
new_window) )
```

---

# F    Results of Additional LLMs

To ensure a fair comparison of positional vectors across various attention mechanisms and positional encodings, we conducted continual pre-training using TinyLlama under consistent settings. However, TinyLlama is a relatively small language model with suboptimal performance and continual pre-training may result in positional vectors exhibiting properties distinct from those obtained through

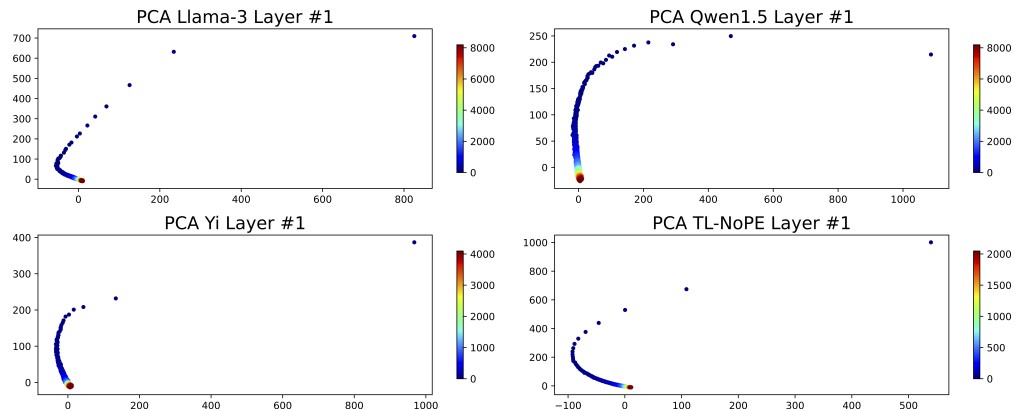

Figure 10: PCA visualization of positional vectors from the 1-st layer of Llama-3-8B, Qwen1.5-7B, Yi-9B, and TL-NoPE-new.

training from scratch. Therefore, we selected three mainstream LLMs: Llama-3-8B [34], Yi-9B [35], and Qwen1.5-7B [36], for comparison. Additionally, we trained a new LLM, TL-NoPE-new, from scratch under the same conditions as TL-NoPE. In a similar vein, we extracted positional vectors using 32K samples from the RedPajama dataset.

### F.1 Formation of Positional Vectors within Context Window

Through principal component analysis (PCA), we first visualize the positional vectors from the initial layer of these LLMs, as illustrated in Figure 10. Consistent with our expectations, the initial tokens exhibit distinct positional information, while the subsequent tokens display a high degree of similarity. This observation supports the conclusion that the first-layer attention mechanism makes the initial tokens form unique positional information, as discussed in Section 3.2.1.

Furthermore, we remove different components of the value vectors at different positions across all attention heads after the first layer. We then evaluate the impact of these modifications on both the positional vectors and the perplexity (PPL). As shown in Table 6, removing the positional basis of the initial tokens significantly degrades the model's performance. Conversely, removing the components from subsequent tokens has a relatively smaller effect, highlighting the pivotal role of initial token positioning in influencing later tokens. However, we observe that larger LLMs are more attuned to semantic information and less affected by the removal of positional vectors compared to smaller LLMs.

Table 6: Results of removing different components in attention.

|  |  | original | w/o value | | w/o positional vector | | w/o positional basis | | w/o semantic basis | |
|---|---|---|---|---|---|---|---|---|---|---|
|  |  | - | 0∼4 | 32-256 | 0∼4 | 32∼256 | 0∼4 | 32-256 | 0∼4 | 32-256 |
| Llama-3 | simi | 1 | 0.75 | 0.9995 | 0.75 | 0.9583 | 0.2059 | 0.9997 | 0.9259 | 0.8666 |
|  | ppl | 6.74 | 16.27 | 6.63 | 17.20 | 8.4 | >1000 | 6.60 | 17.6 | 15.18 |
| Yi-9B | simi | 1 | 0.98 | 0.9999 | 0.92 | 0.9998 | 0.5368 | 1 | 0.91 | 0.9996 |
|  | ppl | 7.08 | 8.03 | 6.56 | 37.92 | 6.62 | >1000 | 6.52 | 42.271 | 7.08 |
| Qwen-1.5-7B | simi | 1 | 0.98 | 0.9997 | 0.9847 | 0.9986 | 0.7382 | 0.9993 | 0.9998 | 0.9951 |
|  | ppl | 7.97 | 9.51 | 8.03 | 9.51 | 8.04 | 217.13 | 7.98 | 8.09 | 8.68 |
| TL-NoPE-new | simi | 1 | 0.70 | 0.95 | 0.69 | 0.95 | 0.41 | 0.93 | 0.99 | 1.0 |
|  | ppl | 11.03 | 224.74 | 22.36 | 263.53 | 20.91 | >1000 | 21.78 | 11.66 | 12.699 |

### F.2 Effect of Positional Vectors on Attention

In line with the experiments conducted in Section 3.2.3, we extract various components from the keys and queries to assess their impact on attention scores. The attention maps for the first 50 tokens are illustrated in Figure 11. When both the positional vectors and positional basis are removed, attention

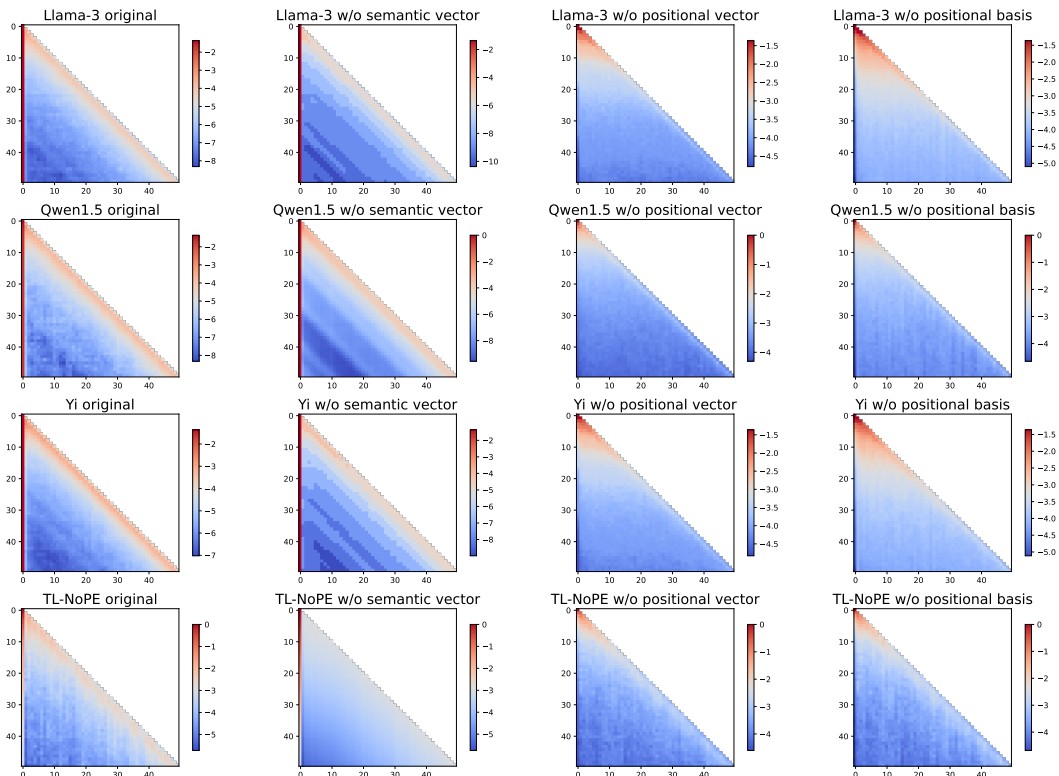

Figure 11: Logarithmic attention maps of Llama-3-8B, Qwen1.5-7B, Yi-9B, and new TL-NoPE.

sinks disappear, and long-term decay is observed, which is consistent with the behavior seen in TinyLlama.

### F.3   Effect of Positional Vectors Beyond Context Window

Table 7: Resuls of PPL and change of positional vectors during direct extrapolation.

| model | context window(C) | PPL(C) | PPL(2C) | Simi(2C) |
|---|---|---|---|---|
| Llama-3-8B | 8192 | 6.74 | >1000 | 0.30 |
| Yi-9B | 4096 | 7.08 | 102.58 | 0.24 |
| TL-NoPE-new | 2048 | 11.75 | 351.49 | 0.71 |

Table 8: Change of attention sinks and output logits beyond context window.

| model | context window (C) | property | 0~C | C~1.5C | 1.5C~2C |
|---|---|---|---|---|---|
| Llama-3-8B | 8192 | attention sink | 0.467 | 0.1 | 0.005 |
| | | logits similarity | 1 | 0.9 | 0.88 |
| Yi-9B | 4096 | attention sink | 0.68 | 0.344 | 0.056 |
| | | logits similarity | 1 | 0.98 | 0.97 |
| TL-NoPE-new | 2048 | attention sink | 0.17 | 0.02 | 0.0006 |
| | | logits similarity | 1 | 0.97 | 0.9 |

We investigate the behavior of positional vectors under direct extrapolation scenarios. Specifically, we evaluate LLMs on sequences twice the length of their maximum context window, comparing the variations in PPL and positional vectors. The results, as shown in Table 7, indicate a sharp

increase in PPL once the context window is exceeded. Additionally, the similarity between positional vectors within and beyond the context window decreases, highlighting the relationship between length extrapolation and the stability of positional representations.

Furthermore, we assess the impact of OOD positional vectors. We analyze the attention scores of initial tokens and the positional vectors of output logits within and beyond the context window. As demonstrated in Table 8, exceeding the context window results in the loss of the attention sinking property, and the positional vectors of the logits exhibit a different distribution compared to those within the context window. This suggests that OOD positional vectors influence the model's distribution. Moreover, OOD positional encodings play a crucial role in large language models (LLMs) that leverage RoPE [15].

### F.4 Experiments of Positional Vector Replacement

To assess whether our positional vector replacement method is specific to TL-NoPE, we applied it to TL-NoPE-new, which was trained from scratch. We evaluated its language modeling performance on the PG-19 dataset, using the same experimental setup as TL-NoPE. As shown in Table 9, our approach effectively extends the context window to 8K tokens and achieves performance comparable to attention scaling.

Table 9: Results of language modeling in PG-19 with TL-NoPE-new

| Interpolation Method | Factor | 2048 | 4096 | 6144 | 8192 |
|---|---|---|---|---|---|
| - | - | 29.0 | 110.7 | 634.8 | 1078.5 |
| Attention Scaling | 1.2 | 29.0 | 27.3 | - | - |
| | 1.5 | 36.8 | 41.6 | 44.4 | 46.2 |
| Positional Vector Replacement(ours) | r=2.5,$\alpha$=0.8 | 32.8 | 29.5 | - | - |
| | r=5, $\alpha$=0.8 | 57.4 | 50.8 | 40.0 | 44.2 |

## G   Visualization of Positional Vectors

We visualize all models listed in Table 1 with PCA. The subsequent figures present the visualization of TL-NoPE, TL-RoPE, TL-ALiBi, TL-Window, TL-Window-RoPE, and TL-Window-80.

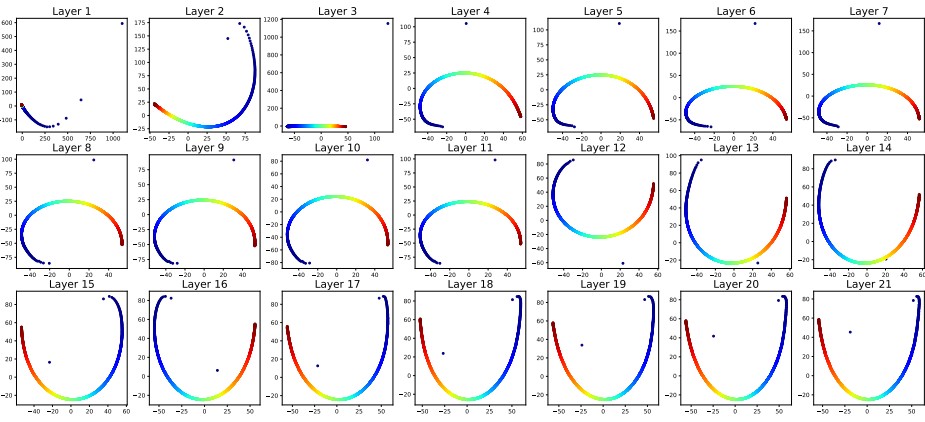

Figure 12: PCA visualization of positional vectors at different layers of TL-NoPE.

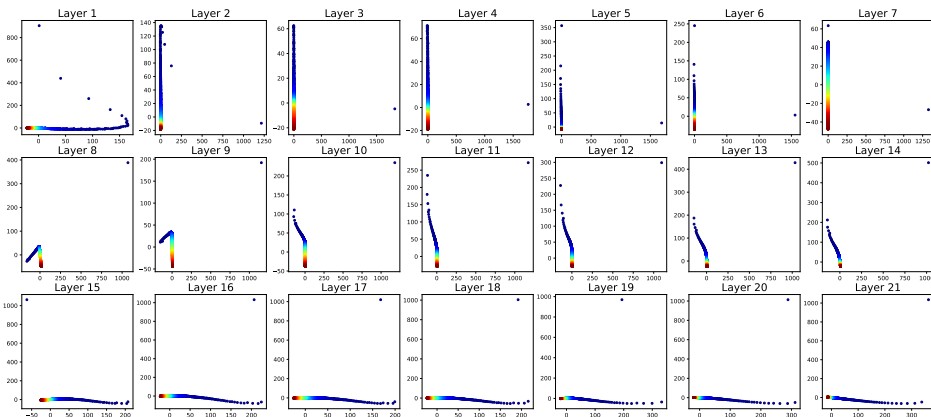

Figure 13: PCA visualization of positional vectors at different layers of TL-RoPE.

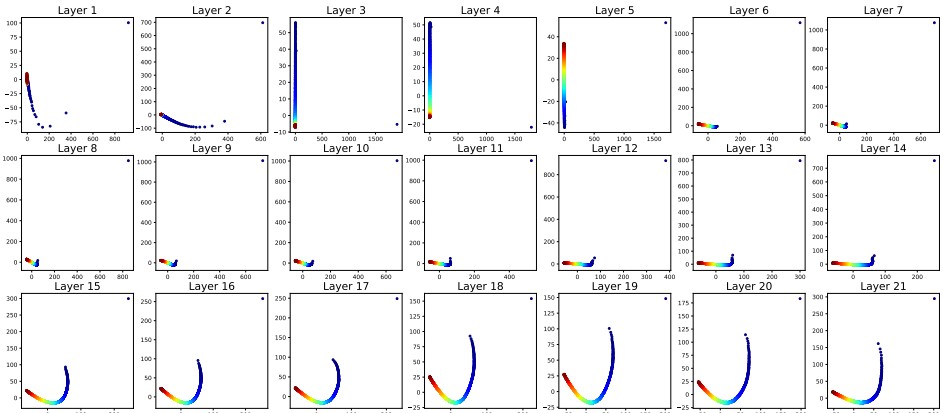

Figure 14: PCA visualization of positional vectors at different layers of TL-ALiBi.

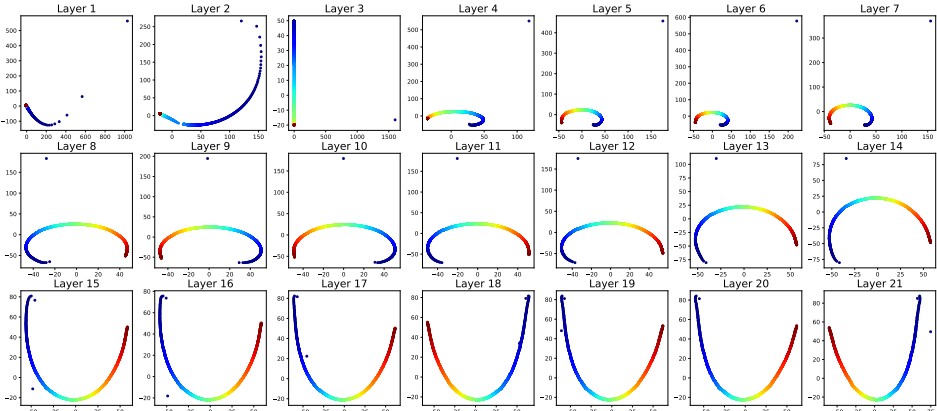

Figure 15: PCA visualization of positional vectors at different layers of TL-Window.

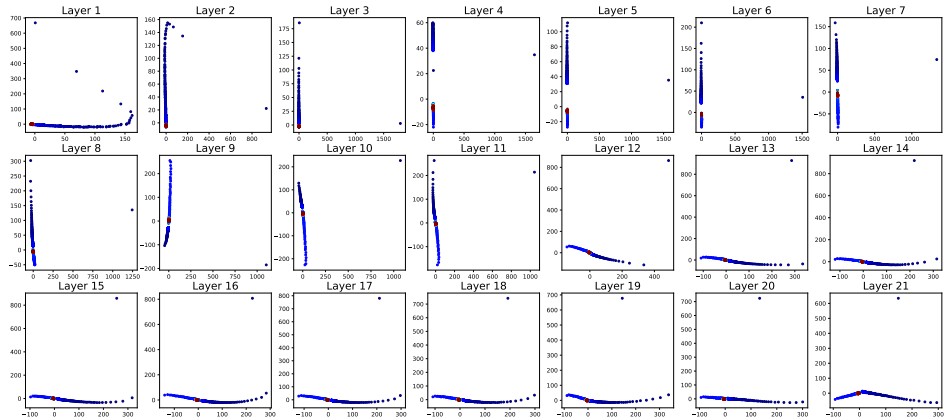

Figure 16: PCA visualization of positional vectors at different layers of TL-Widow-RoPE.

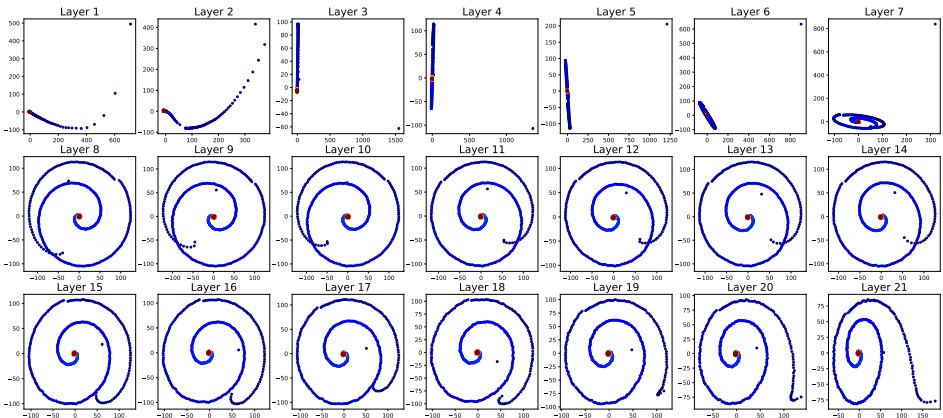

Figure 17: PCA visualization of positional vectors at different layers of TL-Window-80.

