# OpenReview forum: "Exploring Context Window of Large Language Models via Decomposed Positional Vectors"
_NeurIPS.cc/2024/Conference — NeurIPS 2024 spotlight_

### Official Review · Reviewer_yDA4 · 2024-07-06

**Soundness:** 2
**Presentation:** 3
**Contribution:** 3
**Rating:** 6
**Confidence:** 4

**Summary:**

This paper disentangles positional vectors from the hidden states of a pretrained Transformer language model to facilitate the understanding of length extrapolation. After a series of analyses, this paper proposes two context extending techniques. Experiments show that the proposed methods lower the perplexity on the task of language modeling.

**Strengths:**

It's always good to have a mechanistic interpretability view of the hidden states of language models. The findings presented in this paper might inspire follow-up work along this direction.

**Weaknesses:**

The experiments presented in the current draft are not convincing enough to me. See questions below.

**Questions:**

1. Instead of continue training from the tinyllama model, I think training models from scratch using the 50B tokens budget will make the results in this paper more convincing. This is because you can get rid of the ripple effect of the originally used rope positional embeddings. Maybe your models were trying to unlearn rope during the continue training stage?
2. Apart from testing perplexity scores on the task of language modeling, I highly recommend the authors adding the experiment of needle in a haystack, otherwise I do not know if the models are really using all the tokens.
3. How do you decide the values of alpha and lambda in section 4.1 and 4.2? In addition, the temperature scaling technique was also used in several other places [1, 2] with explanations of how they did temperature selection.

[1] YaRN: Efficient Context Window Extension of Large Language Models
[2] Attention Alignment and Flexible Positional Embeddings Improve Transformer Length Extrapolation

**Limitations:**

See above.

---

> ### Author Rebuttal · Authors · 2024-08-06
>
> Thank you for your insightful comments!
> # Q1: Training from Scratch
>
> We initially performed from-scratch pretraining on smaller models and found that the properties of the positional vectors were largely similar to continually-trained models, but the models trained from scratch had inferior performance. To address your question, we pretrained TinyLlama from scratch with the same configuration, but using different positional encodings and attention patterns. Due to time and resource constraints, we only trained TL-NoPE-new for 50B tokens, while the other configurations were trained for only 5B tokens.
>
> * Formation of positional vectors for TL-NoPE: For the TL-NoPE-new trained from scratch, the formation of its positional vectors is similar to the continually trained one: the initial tokens exhibit distinct positional information after the first layer (as shown in Figure 1 in the PDF). As the table below shows, removing the positional information of the initial tokens significantly harms the performance, indicating that the flow of this positional information facilitates the formation of subsequent tokens' positional information.
>
> | |   | original | w/o value | | w/o positional vector |       | w/o positional basis  |       | w/o semantic basis |        |
> |----|----|----|----|-----|--|-----|-----|-----|--------|-----|
> | position | - | - | 0-4 | 32-256 | 0-4 | 32-256 | 0-4  | 32-256 | 0-4 | 32-256 |
> | TL-NoPE-new | simi |     1    |    0.70    |  0.95 | 0.69         |  0.95 |          0.41         |  0.93 |        0.99        |   1.0  |
> | TL-NoPE-new |  ppl |   11.03  |   224.74  | 22.36 |         263.53        | 20.91 |         >1000         | 21.78 |        11.66       | 12.699 |
>
> * Formation of positional vector for TL-NoPE-Window-new: For window attention with NoPE, the positional information flow from initial tokens to subsequent tokens also occurs, and the distinct positional vectors gradually propagate across both windows and layers. The distinct positional vectors of the initial layers are shown in the table below.
>
> | layer |  1 |   2 |   3 |    4 |    5 |    6 |    7 |
> |--|---:|----:|----:|-----:|-----:|-----:|-----:|
> | Distinct Positional Vectors | 79 | 561 | 922 | 1303 | 1682 | 2038 | 2048 |
>
> * Effect of positional vectors on attention: we subsequently remove different components from the query and keys (shown in Figure 2 in PDF), and observe that after removing the positional vectors or positional basis, the long-term decay and attention sink all disappear.
> * Positional vectors change beyond the context window:  as shown in the following table, with inputs exceeding the context window (2K), only models with consistent positional vectors exhibit length extrapolation capacities. In addition, when we employ attention scaling on TL-NoPE, its PPL score on 8K length is 14.05, slightly larger than 11.75 in the 2K length of the original model. The similarity between the original and extended models is 0.97, indicating the interpolation of positional vectors.
>
> | models                | PPL(2K) | PPL(8K) | similarity(8K) |
> |---|---|---|---|
> | TL-NoPE-new  |   11.75 |  351.49 |     0.71 |
> | TL-NoPE-Window-new    | 70.05 |  105.42 |   0.80 |
> | TL-RoPE-Window-new    |  24.95 |   23.43 |           0.98 |
> | TL-NoPE-Window-80-new |  69.91 |   68.12 |           0.99 |
> | TL-ALiBi-new          |        37.63 |   36.17 |           0.99 |
>
> * Effect of OOD Positional Vectors: similarly, the attention sinks and logits of positional vectors of TL-NoPE-new also change after exceeding the context window.
>
> | model       | context window | property  | 2048 | 4096 |   8192 |
> |--|---|----|-----:|-----:|----:|
> | TL-NoPE-new |           2048 | attention sink    | 0.17 | 0.02 | 0.0006 |
> | TL-NoPE-new |           2048 | logits similarity |    1 | 0.97 |    0.9 |
>
> * Proposed Methods: Finally, we test our proposed methods with the from-scratch trained TL-NoPE-new and TL-NoPE-window-new, and observe similar phenomenons. Both positional vector replacement and attention window extension can achieve context window extension. However, due to the properties of these models, the hyper-parameters and the original performance are different.
>
> | Model              | Interpolation Method                | Factor            |  2048 |  4096 |  6144 |   8192 |
> |---|---|---|---:|---:|---:|---:|
> | TL-NoPE-new | -  | -  |  29.0 | 110.7 | 634.8 | 1078.5 |
> | | Attention Scaling   |  lambda = 1.2 |  29.0 |  27.3 | -     | -      |
> | |   |lambda=1.5 |  36.8 |  41.6 |  44.4 |   46.2 |
> | | Positional Vector Replacement(ours) | r=2.5, alpha=0.8 |  32.8 |  29.5 | -     | -      |
> | |  | r=5, alpha=0.8   |  57.4 |  50.8 |  40.0 |   44.2 |
> | TL-NoPE-Window-new | - | -  | 112.7 | 141.5 | 149.7 |  149.7 |
> |  | Attention Window Extension(ours)    | r=2,lambda = 1.1 | 116.5 | 114.7 | 139.6 |  151.7 |
> |   |  | r=4,lambda = 1.2 | 122.3 | 124.4 | 123.3 |  126.0 |
>
> # Q2: Absence of Needle in the Haystack
>
> We initially explored the "needle in a haystack" task. However, due to the poor performance of TinyLlama, the models we continued to pre-train exhibited diminished retrieval capabilities. In our tests, even with an input text length of 200 tokens, the models failed to generate accurate responses, regardless of the positional embeddings and attention patterns employed. Given the difficulty of the "needle in a haystack" task, we did not include the outputs in our paper.
>
> # Q3: How to Decide Values of Alpha and Lambda
>
> For the alpha value in positional vector replacement, we conducted experiments in Appendix E according to the effective interpolation ratio and PPL. We tested different combinations and identified the optimal interpolation ratio, replacement layer, and interpolation times (alpha). For lambda in the attention window extension, we similarly used a PPL-based search to determine the scaling factor. We will further point out how to decide these hyper-parameters and add citations in the final version of our paper.

---

> > ### Comment · Reviewer_yDA4 · 2024-08-10
> > **Thank you for the rebuttal**
> >
> > Q1 and Q3: Thank you for addressing the concerns. We are good now.
> >
> > Q2: I believe there should be other methods to test whether the model is truly utilizing long-context information. I would assume TinyLLama still has a decent in-context learning ability given its size. If so, one experiment I can suggest is:
> >
> > Take a segment (S) of natural text with length L.
> > Repeat it N times and concatenate them to form an artificial sequence (A) with a length of L*N.
> > Feed A into your length-extended model and observe the output.
> > If the model learns to use long-context information, it should output the first token of the original short sequence (S[0]). Note that S[0] is not a natural continuation of A; the model will only output S[0] if it leverages the long artificial in-context examples provided. You can then experiment with different values of L and N to demonstrate the model's ability to process long-context information.
> >
> > The experiment above is just one example. Feel free to devise other methods that can effectively demonstrate the long-context processing capabilities of your model.

---

> ### Author Response · Authors · 2024-08-11
> **Our methods can keep the long context utilization capactity of LLMs to some extent.**
>
> Thank you for your response and valuable suggestion!
>
> To assess the long-context utilization capability, we have adopted your recommendation and evaluated the LLMs using ICL. Specifically, we randomly sampled consecutive substrings from the RedPajama dataset, with lengths of 20, 50, and 200 tokens, respectively. These substrings were then repeated to achieve varying lengths, and we evaluated the accuracy of the models' ability to repeat the first tokens of substrings correctly. The results are presented in the following table.
> | Total Length(L*N) | -  |      | 1K   |      |      | 2K   |      |      | 4K   |      |      | 8K   |      |
> |-------|----|------|------|------|------|------|------|------|------|------|------|------|------|
> | Text Length(L)    | -                                               |   20 |   50 |  200 |   20 |   50 |  200 |   20 |   50 |  200 |   20 |   50 |  200 |
> | TL-NoPE-Window    | original                                        | 0.98 | 0.98 | 0.92 | 0.95 | 0.97 | 0.93 | 0.08 | 0.07 | 0.05 | 0.08 | 0.07 | 0.05 |
> |                   | attention window extension(r=2, lambda=1.1)     | 0.99 | 0.96 |  0.9 | 0.89 | 0.89 | 0.92 | 0.66 | 0.89 | 0.92 | 0.06 | 0.07 | 0.06 |
> |                   | attention window extension(r=4, lambda=1.2)     | 0.88 | 0.74 |  0.9 | 0.75 |  0.7 |  0.9 | 0.48 |  0.7 | 0.83 | 0.44 | 0.59 | 0.74 |
> | TL-NoPE-new           | original                                        |    1 |  0.9 | 0.86 |    1 |  0.9 | 0.86 | 0.11 | 0.49 | 0.13 | 0.06 | 0.07 | 0.01 |
> |                   | positional vector replacement(r=2.5, alpha=0.8) | 0.51 | 0.66 | 0.74 | 0.46 | 0.68 | 0.73 | 0.69 | 0.82 | 0.74 | -    | -    | -    |
> |                   | positional vector replacement(r=5, alpha=0.8    |  0.3 | 0.26 | 0.23 | 0.25 | 0.21 | 0.28 | 0.24 | 0.22 | 0.29 | 0.21 | 0.49 | 0.64 |
>
> After exceeding the context window, the performance of models deteriorates significantly, whereas our methods maintain a degree of capability to utilize longer contexts effectively. Specifically, for TL-Window-NoPE, as the substring length increases, the model exhibits slower degradation of performances. For TL-NoPE, with much longer substrings (for example, 200 tokens), the model even demonstrates superior performance at 8K tokens compared to 2K tokens. These enhancements underscore that our models can leverage longer contexts for enhanced performance.

---

> > ### Comment · Reviewer_yDA4 · 2024-08-11
> > **Thank you. I have one more question.**
> >
> > Thank you for the additional experiments. One question: Could you also report the numbers of TL_NoPE with your proposed positional vector replacement method?

---

> > > ### Author Response · Authors · 2024-08-11
> > > **Experiments with TL-NoPE.**
> > >
> > > Thank you for your response! We have evaluated the continually trained TL-NoPE and observed similar phenomena, with the hyperparameters listed in the second column of the table below. As demonstrated in the table, TL-NoPE with our method can utilize the long context to some extent to repeat the first token. Moreover, longer substrings (200 tokens) enhance the model's performance in in-context learning (ICL) with long contexts, which underscores the effectiveness of our methods in leveraging long contexts. Finally, further expanding the interpolation ratio and times can yield better performance on longer inputs, albeit with a slight degradation in performance on shorter inputs.
> > >
> > > | Total Length(L*N) | -                                             |      | 1K   |      |      | 2K   |      |      | 4K   |      |      | 8K   |      |
> > > |-------------------|-----------------------------------------------|------|------|------|------|------|------|------|------|------|------|------|------|
> > > | Text Length(L)    | -                                             |   20 |   50 |  200 |   20 |   50 |  200 |   20 |   50 |  200 |   20 |   50 |  200 |
> > > |      TL-NoPE      |                    original                   |   1  | 0.99 | 0.94 |   1  | 0.99 | 0.92 | 0.11 | 0.05 | 0.06 | 0.06 | 0.06 | 0.06 |
> > > |                   | positional vector replacement(r=2, alpha=1.1) | 0.97 | 0.97 | 0.91 | 0.95 | 0.95 | 0.93 | 0.67 | 0.74 | 0.68 |   -  |   -  |   -  |
> > > |                   | positional vector replacement(r=5, alpha=1.3) | 0.65 | 0.85 | 0.86 | 0.56 |  0.8 | 0.87 |  0.5 | 0.72 | 0.84 | 0.14 | 0.28 | 0.68 |
> > > |                   | positional vector replacement(r=6, alpha=1.4) | 0.61 | 0.85 | 0.83 | 0.46 | 0.68 | 0.86 | 0.43 | 0.59 | 0.77 | 0.23 | 0.43 | 0.73 |

---

> > > > ### Comment · Reviewer_yDA4 · 2024-08-11
> > > > **Thank you for the additional experiments**
> > > >
> > > > Thank you for the additional results. It would be great if you can add them to the final revision.
> > > >
> > > > In addition, please also add the missing references [1, 2] to the final revision so that the readers are aware of the similar temperature scaling technique explored in the context of other positional embeddings.
> > > >
> > > > I increased the score to 6.

---

### Official Review · Reviewer_1sVW · 2024-07-08

**Soundness:** 3
**Presentation:** 4
**Contribution:** 3
**Rating:** 7
**Confidence:** 3

**Summary:**

This paper proposes a mean-based decomposition technique to analyze the formation and effect of positional encodings in LLMs. It then uses these results to propose methods to extend the context window, resulting in models that generalize better to longer texts.

**Strengths:**

1. This paper is very well-written, and the main findings are properly highlighted.

2. This paper not only explains how positional vectors are formed, but also introduces methods to interpolate them based on the findings.

3. Experiments are performed to show that the new methods result in better perplexity scores beyond the context window.

**Weaknesses:**

I believe this contribution is novel and insightful enough, and there is no apparent weakness.

**Questions:**

1. The legends and graphs in Figure 4 overlap.

**Limitations:**

The authors have adequately addressed the limitations of their work.

---

> ### Author Rebuttal · Authors · 2024-08-06
>
> Thank you for your helpful comments! We will revise Figure 4 in the final version.

---

> > ### Comment · Reviewer_1sVW · 2024-08-13
> >
> > Thanks for the response!

---

### Official Review · Reviewer_R24u · 2024-07-17

**Soundness:** 4
**Presentation:** 3
**Contribution:** 4
**Rating:** 7
**Confidence:** 4

**Summary:**

This paper dives into the inner workings of how transformer-based language models handle positional information. By decomposing hidden states into semantic and positional vectors, the authors give a series of analysis about how the positional information are encoded and propagated through layers. I believe this work offers valuable insights for understanding the positional information within the transformer architecture.

**Strengths:**

Very detailed and clear analysis about how the positional information is encoded and propagated within the transformer architecture, and to the best of my knowledge, I have not seen similar work before. I particularly enjoyed reading Figure 2 and 3, which shows how positional information is propagated through layers and goes beyond the window size, and shows how the manipulation of the positional embedding causally influence the attention patterns, particularly removing the attention sink.

**Weaknesses:**

There are few points that I would like to suggest here to make the paper even stronger.

- Section 4 feels weak and unnecessary. The performance of replacing the positional vector, if my understanding is correct, seems to be much worse than Dynamic NTK. Given the current mainstream approach is modifying the base of Rope (like YaRN), which is much easier than the approach proposed by this work, I do not think this work’s proposed context extension will be accepted by mainstream model builder.
- That being said, I think the in-depth analysis of the positional embeddings are strong enough for me to give an acceptance (I learned a lot from it), so **I would strongly suggest removing the content of section 4, and use its space for more experimental analysis of the positional vectors**

There are a few important problems that I believe will receive the communities’ attention and worth being addressed:

- Although this paper shows the positional information can propagate through layers (Figure 2), in practice, many work found that models with window attention cannot pass the needle in a haystack test, and this is why Mistral 1.5 changed its attention back to full attention. It would be insightful if the authors can discuss the relationships between positional information and needle-in-a-haystack performance (because needle in haystack is what makes long-context models useful), i.e., why window attention cannot pass needle in haystack even it does have the correct positional information?
- This paper’s analysis is restricted on TinyLLaMA, but TinyLLaMA is not a widely used open-source model, thus casting the doubt whether this discovery of this paper will hold for other model families, particularly mainstream open-weight models like LLaMA 3, Mistral, QWen or Yi. I would strongly suggest the authors verify the behavior of positional embedding on either LLaMA 3, Mistral, QWen, or Yi.

Currently I’m given a borderline accept, and I will consider increasing my scores if the authors could either (1) discuss the relationship between positional vectors v.s. needle-in-a-haystack or (2) verify that the properties of positional vectors hold for LLaMA 3, Mistral, QWen or Yi (any 2 out of the 4).

**Questions:**

see the above weakness section

**Limitations:**

see the above weakness section

---

> ### Author Rebuttal · Authors · 2024-08-06
>
> Thank you for your insightful comments!
>
> # W1: Unnecessaries of Section 4
>
> In Section 4, the proposed methods are significant evidence for our analysis of the relationship between positional vectors and the context window. Our experiments substantiate our previous viewpoints. For instance, interpolating positional vectors can extend the context window to some extent. Although it has poorer performance than Dynamic NTK, it provides a new approach to solving the problem of context window extension and can be applied to models without RoPE. In the final version with one additional page, we will adopt your suggestions by supplementing more experiments and condensing the content of this section to some extent.
>
> # W2: Failure of Window Attentions in Needle in the Haystack
>
> Why can window attention hardly solve the needle in the haystack problem?
>
> * Some work has reported that models with full attention can utilize "retrieval heads" to directly retrieve critical information from the needle [1], while window attention cannot directly attend to these tokens.
>
> * Models with window attention acquire knowledge only through the indirect transmission of information between windows. However, previous work has highlighted that there is severe attenuation in the information passing between windows, making it difficult to leverage semantic information from distant tokens [2].
>
> * Positional information tends to be a kind of global information at a given position, ensuring the model can produce coherent output, and is orthogonal to the semantic information of the input.
>
> To validate this, we first compare the hidden states of tokens when using the full input sequence versus using only the window-sized input at a time. We test TL-RoPE-Window and Mistral-7B-v0.1 with input lengths of 16K and 32K, respectively. We find that the similarity of the last layer's hidden states under both settings was above 0.98, suggesting that information from tokens outside the window has little impact.
>
> Additionally, we test the scenario where the "needle" is placed at the beginning but is modified into random tokens from the vocabulary. The hidden states of the outputs remain largely unchanged, further supporting our hypothesis.
>
> [1] Retrieval Head Mechanistically Explains Long-Context Factuality
>
> [2] Dissecting Transformer Length Extrapolation via the Lens of Receptive Field Analysis
> # W3: Absence of Some Mainstream LLMs
> Considering resource constraints, we only continuously pre-trained TinyLlama with modified positional encoding under identical settings for a fair comparison. The mainstream models are all based on RoPE, and continuing pre-training is prohibitively expensive; hence, we only examined the positional vectors of the original Llama3-8B, Yi-9B, and Qwen1.5-7B.
>
> * Formation of positional vectors: After the first layer, the initial positions have significantly different positional vectors compared to other tokens, as shown in Figure 1 of the PDF. Furthermore, we remove different components from values at different positions and observe changes in the positional vectors and PPL, as shown in the table below. The findings are consistent with the original paper, indicating that the initial tokens play a critical role in shaping the positional vectors and influencing the model’s PPL. When the positional information of the initial tokens is lost, the model loses its ability to produce coherent output. Moreover, Llama-3-8B and Yi-9B are more sensitive to semantic information than smaller models.
> |  |  | original | w/o value || w/o positional vector | | w/o positional basis  | | w/o semantic basis | |
> |---|---|---|---|---|---|---|---|---|---|---|
> | position | - | - | 0-4 | 32-256 | 0-4 | 32-256 | 0-4  | 32-256 | 0-4 | 32-256 |
> | Llama-3 | simi | 1 | 0.75 | 1.0 | 0.75 | 0.96 | 0.21 | 1.0 | 0.93 | 0.87 |
> | Llama-3 | ppl | 6.74 | 16.27 | 6.63   | 17.20 | 8.4 | >1000 | 6.60 | 17.6 | 15.18 |
> | Yi-9B | simi |1| 0.98 | 1.0 | 0.92  | 1.0 | 0.54 | 1 | 0.91 | 1.0 |
> | Yi-9B | ppl | 6.51 | 8.03 | 6.56 | 37.92 | 6.62 | >1000  | 6.52 | 42.27 | 7.08 |
> | Qwen1.5-7B | simi | 1 | 0.98 | 1.0 | 0.98 | 1.0 | 0.74 | 1.0 | 1.0  | 0.99 |
> | Qwen1.5-7B | ppl | 7.97| 9.51| 8.03 | 9.51 | 8.04 | 217.13 | 7.98 | 8.09 | 8.68 |
>
> * Impact of Positional Vectors on Attention: We examine the changes in attention after removing the positional vectors, as illustrated in Figure 2 of the PDF. We observe that attention sinks and long-term decay are eliminated after removing the positional vectors or basis.
>
> * Effect of Positional Vectors Beyond the Context Window: As shown in the following table, after exceeding the context window, models without an extended context window experience a sharp change in positional vectors and PPL, similar to TL-RoPE. By using dynamic NTK, we achieve interpolation of the positional vectors, maintaining high similarity to the original positional vectors.
> | model | context window(W) | PPL(W) | PPL(2W) | Simi(2W) | PPL(NTK, 2W) | Simi(NTK, 2W) |
> |---|---|---|---|---|---|---|
> | Llama-3-8B | 8192 | 6.74 | >1000 | 0.30 |7.84 | 0.96 |
> | Yi-9B | 4096 | 6.51 | 102.58 | 0.24 | 6.70 | 0.99 |
> * Effect of OOD Positional Vectors: After exceeding the context window, the properties of high attention scores on the first token (attention sinks) in Llama-3 and Yi-9B rapidly disappear, and the logits of the positional vectors differ from those within the window, as shown in the table below. However, the change in logits for Yi is not obvious, although it has formed the same pattern as shown in Figure 5 (right) of the paper.
> | model | context window (W) | property | 0~W   | W~1.5W | 1.5W~2W |
> |---|---|---|---|---|---|
> | Llama-3-8B | 8192 | attention sink    | 0.47 | 0.1 | 0.005 |
> | Llama-3-8B | 8192 | logits similarity | 1 | 0.9 | 0.88 |
> | Yi-9B | 4096 | attention sink | 0.68 | 0.34 | 0.06 |
> | Yi-9B | 4096 | logits similarity | 1 | 0.98 | 0.97 |

---

### Author Rebuttal · Authors · 2024-08-06

Thank you for your insightful comments! The supplementary PDF includes figures that support our rebuttals.

---

### Author Response · Authors · 2024-08-10

Hi Reviewers,

Can I please know whether our response addresses your questions? We are curious if there are any particular limitations or weaknesses that prevent you from raising your score. If so, we'd like to have the opportunity to address those during this discussion period.

---

### Decision · Program_Chairs · 2024-09-25

**Decision:**

Accept (spotlight)

**Comment:**

This paper presents a novel approach to understanding how positional information is encoded and propagated within transformer-based language models. With a mean-based decomposition, they decompose hidden states into semantic and positional vectors, the authors provide detailed insights into the role and behavior of positional embeddings. Also, the work proposes methods for extending the context window based on these findings, aiming to improve model performance on longer sequences.

Pros:

The reviewers mostly recognize the paper's in-depth analysis of how positional information is encoded and propagated within transformer layers. This is a novel contribution that can benefit both research and industry aspects of LLMs.

The paper is well-written and organized, with the main findings and contributions clearly highlighted. The figures are praised for their clarity and the insights they provide into the positional information.

Cons:

Generalization Across Models: Both Reviewer R24u and yDA4 highlighted concerns regarding the generalizability of the findings. The experiments were conducted on TinyLLaMA, which is a less powerful model, raising doubts about whether the results would hold for more mainstream models.